# Identification of 3-chymotrypsin like protease (3CLPro) inhibitors as potential anti-SARS-CoV-2 agents

Vicky Mody[1,3], Joanna Ho[1], Savannah Wills[1], Ahmed Mawri[1], Latasha Lawson[1], Maximilian C. C. J. C. Ebert[2], Guillaume M. Fortin[2], Srujana Rayalam[1] & Shashidharamurthy Taval [1,3 ✉]

Emerging outbreak of severe acute respiratory syndrome coronavirus-2 (SARS-CoV-2) infection is a major threat to public health. The morbidity is increasing due to lack of SARS-CoV-2 specific drugs. Herein, we have identified potential drugs that target the 3-chymotrypsin like protease (3CLpro), the main protease that is pivotal for the replication of SARS-CoV-2. Computational molecular modeling was used to screen 3987 FDA approved drugs, and 47 drugs were selected to study their inhibitory effects on SARS-CoV-2 specific 3CLpro enzyme in vitro. Our results indicate that boceprevir, ombitasvir, paritaprevir, tipranavir, ivermectin, and micafungin exhibited inhibitory effect towards 3CLpro enzymatic activity. The 100 ns molecular dynamics simulation studies showed that ivermectin may require homodimeric form of 3CLpro enzyme for its inhibitory activity. In summary, these molecules could be useful to develop highly specific therapeutically viable drugs to inhibit the SARS-CoV-2 replication either alone or in combination with drugs specific for other SARS-CoV-2 viral targets.

[1] Department of Pharmaceutical Sciences, School of Pharmacy, Philadelphia College of Osteopathic Medicine – Georgia Campus, Suwanee, GA, USA. [2] Chemical Computing Group, 910-1010 Sherbrooke W, Montreal, QC H3A 2R7, Canada. [3] These authors contributed equally: Vicky Mody, Shashidharamurthy Taval. ✉email: rangaiahsh@pcom.edu

The major pandemic outbreak of the 21st century due to severe acute respiratory syndrome coronavirus-2 (SARS-CoV-2) has become a global threat to public health because of its high rate of infection leading to mortality. As of 24 December 2020, there are a total of 13,881,620 COVID-19 positive cases and 272,820 deaths in the United States alone and 64,326,880 confirmed cases and 1,488,992 deaths globally (https://coronavirus.jhu.edu/). The death toll is increasing at an alarming rate because of the lack of COVID-19 specific drugs or vaccines. Development, validation, and approval of COVID-19 specific drugs takes years[1]. Therefore, the idea of drug repositioning, also known as repurposing, is an important strategy to control the sudden outbreak of life-threatening infectious agents that spread rapidly. FDA approved anti-viral drugs are known to be safe for use in humans[2], but their effectiveness against SARS-CoV-2 needs to be experimentally validated. Several FDA approved anti-viral drugs such as favipiravir, danoprevir, darunavir, lopinavir, oseltamivir, ritonavir, remdesivir, and umifenovir are in clinical trials to study anti-COVID-19 activity[3]. However, the effectiveness of these drugs for preventing or reducing the severity of symptoms of COVID-19 has not yet been completely established. Therefore, there is an urgent need to identify additional drug candidates to target different SARS-CoV-2 proteins for enhanced efficacy in the treatment of COVID-19.

Recently, Wu et al.[4] sequenced and compared SARS-CoV-2 genome with other coronaviruses (CoVs) and confirmed that novel SARS-CoV-2 belongs to β-CoVs, which were originally found in bats and have now adapted to infect humans. CoVs are RNA viruses with positive-sense single stranded RNA (+ssRNA) as their genetic material[5] and recent studies have shown that SARS-CoV-2 shares ~89% sequence similarity with other SARS-CoVs[4]. Additionally, SARS-CoV-2 has similar genetic organization as other SARS-CoVs with a 5′-untranslated region followed by 16 non-structural proteins (open reading frame; ORF1a and ORF1b complex) also called as replicase complex, and the structural proteins such as spike (S), envelop (E), membrane (M), and nucleocaspid (N) protein along with other accessory proteins present towards the 3′ end[6]. The life cycle of the virus begins with the binding of the S protein of the virus to its receptor on the host cells, the ACE2. The binding is then followed by the fusion of the viral envelop with host cell membrane and the release of the viral genome into the cytoplasm[7]. The viral genome (+ssRNA) hijacks the host ribosomes and gets translated into ~ 800KDa large polypeptide (PP) chain. The newly generated PP chain is auto-proteolytically cleaved by two proteases such as papain like proteases (PLpro) and 3-chyomotrypsin like protease (3CLpro), encoded by the viral genome, to generate several non-structural proteins (NSPs) required for the viral replication. 3CLpro is also called the main protease (Mpro) and plays a major role in the viral replication. PLpro and 3CLpro cleaves the PP chain into 16 NSPs and out of the 16 NSPs generated, 11 NSPs are generated by the 3CLpro, making this protease one of the major targets for developing anti-SARS-CoV drugs[8,9]. On the other hand, structural and other accessary proteins are generated through a unique mechanism called sub-genome (Sg) translation. Sg's are produced through discontinuous transcription from 5′ end of the anti-sense viral RNA[10–13]. After successful genome replication and translation, NSPs, structural proteins and accessory proteins assemble along with positive-sense viral RNA genome to form a new virion. The CoVs genome and proteolytic cleavages by PLpro and 3CLpro is illustrated in Fig. 1.

The genomic and protein sequences for SARS-CoV-2 are publicly available from the NIH gene data bank[4,6,14]. Herein, we have selected 3CLpro of SARS-CoV-2 as a target to identify potential inhibitors since this protease is indispensable for viral replication and hence an excellent drug target[9]. The structure of 3CLpro protein of SARS-CoV-2 in complex with an inhibitor N3 is available in the PDB database (ID: 6LU7). To identify the FDA approved drugs as inhibitors for 3CLpro, in silico drug screening studies were carried out. In all, 3987 FDA approved drugs (SuperDrugs2 database) were sorted as viral protease inhibitors (PIs), viral non-protease inhibitors (VNIs) and off-target drugs (OTDs), and screened for the anti-3CLpro activity using the Molecular Operating Environment (MOE) software. The protein structure-based drug design using computational methods is an alternative for screening of currently approved drugs to rapidly identify potential drug candidates for the treatment of emerging infectious diseases such as COVID-19[15–17]. However, the potential for false positives with computational modeling is one of the most common limitation of docking studies[18]. Therefore, we have established SARS-CoV-2 3CLpro enzymatic assays for selected drugs using commercially available 3CLpro protease inhibitor screening assay kits to evaluate the in vitro inhibitory activity of the drugs and investigated whether any correlation exist between the computational binding score and the in vitro inhibitory activity. In this report, we have selected 47 from the list of 3987 FDA approved drugs based on binding score, side effects, half-life, active form, immunosuppressant properties, autofluorescence, and availability for an in vitro 3CLpro enzymatic inhibitor-screening assay. We observed that, boceprevir, ombitasvir, paritaprevir, tipranavir, and micafungin exhibited partial inhibitory effect whereas, ivermectin blocked more than 85% of 3CLpro activity of SARS-CoV-2. Although the anti-viral activity of ivermectin mediated through the blocking of α/β1 importin[19–23] is established, herein we report the inhibitory effects of ivermectin on 3CLpro enzyme of SARS-CoV-2, suggesting additional anti-viral mechanism of ivermectin towards SARS-CoV-2.

## Results

**In silico screening of FDA approved drugs for potential binding to SARS-CoV-2 3CLpro enzyme.** All the 3987 FDA approved drugs (downloaded from SuperDrugs2) were sorted as PIs, VNIs, and OTDs and docked with monomeric form of 3CLpro protein using the Molecular Operating Environment (MOE) software. Drugs were ranked according to the highest binding energy (S score). S score was calculated using the London dG score for placement and GBVI/WSA dG score for refinement of poses. A binding energy with ≤−6.5 kcal/mol (S score) was selected as a cutoff to rank the drugs with highest binding affinity[16]. The list was further narrowed down based on the criteria listed in the Methods section. We found that 56 drugs have a binding affinity of <−6.50 kcal/mol for the 3CLpro enzyme (Table 1). The computational study suggested that the list of drugs identified might inhibit the SARS-CoV-2 viral replication by targeting the viral 3CL protease. However, the potential for false positives with the predicted binding score is one of the most common limitation of docking studies[24,25]. Hence, to rule out any ambiguity in false prediction, we carried out the in vitro enzymatic assay to identify potential therapeutics and investigate correlation between the binding score and the in vitro activity. In the current study, we did not find any correlation between the in vitro results of selected drugs and their computational inhibition constants. Even though, computational studies are being widely used to predict the initial protein-drug interactions, in vitro screening of the drugs is necessary to confirm the inhibitory activities of the drugs.

**Inhibition of SARS-CoV-2 3CLpro enzymatic activity by selected drug candidates.** To further validate the SARS-CoV-2 3CLpro inhibitory activity of the selected drugs from computational studies, we performed an in vitro enzymatic inhibitory

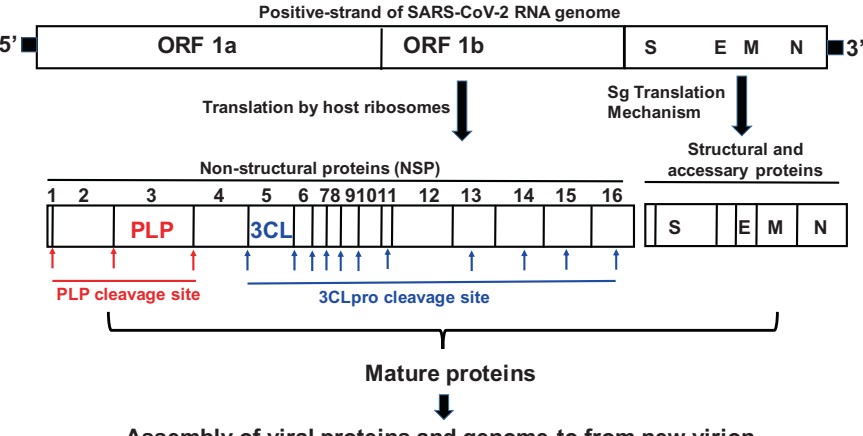

**Fig. 1 Representation of SARS-CoV-2 genome arrangement and protease cleavage sites.** Genetic material (RNA) of SARS-CoV-2 enters the host cell and borrows ribosome to translate into 16 nonstructural proteins upon auto-proteolytic cleavage by PLpro and 3CLpro enzymes. Structural proteins (S, E, M, and N) are translated by sg translation mechanism. All mature proteins assemble along with positive-sense single stranded RNA genome to form new virion. Arrows indicates the protease cleavage sites. ORF open reading frame, PLP papain like protease, 3CLP 3-chymotrypsin like protease, S spike protein, E envelop protein, M membrane protein, N nucleocaspid protein.

assay using commercially available assay kits. The background autofluorescence of the selected 56 compounds was measured by excitation/emission wavelength of 360/460 nm. Nine out of 56 drugs exhibited extremely high autofluorescence (Supplementary Fig. 2) and were therefore eliminated from the in vitro protease inhibitor enzymatic assay. The remaining 47 drugs were sorted as PI, VNIs, and OTDs. The specific mechanism of action and their clinical uses are listed in Supplementary Table 1. The 3CLpro inhibitory activity was screened at 50 μM concentration of the drugs. Among the 17 PIs screened, boceprevir, paritaprevir and tipranavir significantly inhibited the 3CLpro enzymatic activity by 45, 70, and 64%, respectively (Fig. 2). Out of the 17 VNIs, only ombitasvir was able to partially block (65%) the 3CLpro enzymatic activity (Fig. 3). Interestingly, as shown in Fig. 4, out of 13 OTDs only ivermectin completely blocked (>80%) the 3CLpro activity at 50 μM concentration. Additionally, micafungin exhibited partial inhibitory activity (57%) against 3CLpro of SARS-CoV-2.

The compounds that exhibited more than 50% inhibitory activity were subjected to subsequent dose-dependent studies to calculate the concentration required to inhibit 50% of the 3CLpro enzymatic activity ($IC_{50}$). Boceprevir, ivermectin, micafungin, ombitasvir, paritaprevir, and tipranavir were subjected to dose-dependent inhibitory activity studies. As shown in the Fig. 5, ivermectin inhibited more than 85% of the enzymatic activity at 50 μM concentration, whereas micafungin and paritaprevir inhibited around 80% of the enzymatic activity at 100 μM concentration. Both tipranavir and ombitasvir were able to inhibit only 50% of the enzymatic activity even at 100 μM concentration (Fig. 5). The percent enzymatic activity versus the log concentration of the inhibitors was used to calculate the $IC_{50}$ values using non-linear curve fit model as described under Methods section. The calculated $IC_{50}$ values for ivermectin, tipranavir, boceprevir, micafungin, paritaprevir, and ombitasvir were found to be 21.5, 27.7, 31.4, 47.6, 73.4, and 75.5 μM, respectively (Table 2). Taken together, these studies suggest that the molecules listed above exhibited inhibitory activity against 3CLpro enzyme of SARS-CoV-2.

**Structural interaction of boceprevir, paritaprevir ombitasvir, tipranavir, ivermectin, and micafungin with the enzymatic active site of 3CLpro.** The presence of an unconventional catalytic cysteine residue in 3CLpro makes it unique as compared to other chymotrypsine like enzymes and other Ser (or Cys) hydrolases[26]. In addition, the 3CLpro consist of a catalytic $Cys^{145}$-$His^{41}$ dyad instead of a canonical Ser(Cys)-His-Asp(Glu) triad[27]. This catalytic dyad is activated in the presence of the substrate containing Leu-Gln↓Ser-Ala-Gly (↓ marks the cleavage site)[26]. Structurally the catalytic residues $Cys^{145}$ and $His^{41}$ in 3CLpro are buried in an active site cavity located on the surface of the protein (Supplementary Fig. 1) and both residues are at a distance of 3.8 Å[26,28]. This distance might be long enough to prevent formation of any intermolecular hydrogen bond between $Cys^{145}$ and $His^{41}$ at physiological pH. Additionally, the $Cys^{145}$ is protonated at physiological pH and $His^{41}$ is present in the neutral state. The substrate binding triggers the intramolecular proton transfer from $Cys^{145}$ to $His^{41}$ triggered via the attack of $Cys^{145}$-Sulfur onto the (C=O) of the substrate peptide bond[27]. During catalytic activity, the protonated $His^{41}$ is stabilized by the presence of $H_2O$ molecule close to $His^{41}$. Thus, the ability to form a hydrogen bond with both $Cys^{145}$ and $His^{41}$ residues or either with one of these residues might be a characteristic of a good inhibitor.

Using computational modeling, the structural interaction of boceprevir, paritaprevir ombitasvir, tipranavir, ivermectin, and micafungin with 3CLpro SARS-CoV-2 enzyme was studied. We found that carbonyl (C=O) group in ivermectin and boceprevir forms a hydrogen bond with $Cys^{145}$ which might explain their inhibitory activity (Fig. 6a, e; panel I and II). Similar type of hydrogen bonding was also observed between carbonyl (C=O) group in paritaprevir next to the sulfonamide and $Cys^{145}$ and also between sulfonamide group in tipranavir and $Cys^{145}$ (Fig. 6b, c; panel I and II). In addition, same carbonyl (C=O) group in paritapavir also shows hydrogen bonding interaction with $His^{41}$ (Fig. 6b; panel I and II). However, ombitasvir and micafungin show a hydrogen bonding interaction with $Glu^{166}$ (Fig. 6d, f; panel I and II). The hydrogen bonding interaction of $Glu^{166}$ with ombitsavir and micafungin is very important, as the $Glu^{166}$ is responsible for the formation of the homodimer of 3CLpro in SARS-CoV-2[28]. This dimer form is important for its enzymatic activity and any interaction with $Glu^{166}$ can result in the formation of an inactive monomer which interferes with the enzyme activity of 3CLpro[28]. Computational modeling illustrations are presented in Fig. 6a–f; panel-III shows the interaction of the drugs with the lipophilic pocket of the 3CLpro enzymatic site.

**Table 1 List of all the drugs with highest S score for 3CL protease calculated by computational studies (S score).**

| SL. No. | Drug | Binding affinity (Kcal/mol) to 3CLpro |
|---|---|---|
| 1 | Abacavir | −6.6178 |
| 2 | Amprenavir | −8.29197 |
| 3 | Asunaprevir | −9.64249 |
| 4 | Atazanavir | −8.51517 |
| 5 | Atrovastatin | −7.98771 |
| 6 | Beclabuvir | −8.45563 |
| 7 | Boceprevir | −8.4209 |
| 8 | Camostat | −7.66508 |
| 9 | Candicidin | −9.05496 |
| 10 | Chloroquine | −6.7514 |
| 11 | Daclatasvir | −8.47852 |
| 12 | Danoprevir | −9.26908 |
| 13 | Darunavir | −8.59002 |
| 14 | Delavirdine (mesylate) | −7.3092 |
| 15 | Elbasvir | −8.77218 |
| 16 | Elvitegravir | −7.48581 |
| 17 | Etravirine | −6.91494 |
| 18 | Favipiravir | −4.23592 |
| 19 | Fumagillin | −7.25118 |
| 20 | Gabexate | −7.24702 |
| 21 | Glecaprevir | −8.70897 |
| 22 | Grazoprevir | −8.85946 |
| 23 | Hydroxychloroquine | −7.03572 |
| 24 | Indinavir | −8.39968 |
| 25 | Itraconazole | −7.99098 |
| 26 | Ivermectin | −7.74053 |
| 27 | Ledipasvir | −9.52825 |
| 28 | Lopinavir | −9.10074 |
| 29 | Maraviroc | −8.19033 |
| 30 | Methylprednisolone | −6.42667 |
| 31 | Micafungin | −9.60339 |
| 32 | Nelfinavir | −8.55575 |
| 33 | Ombitasvir | −8.96571 |
| 34 | Oseltamvir Phosphate | −7.04549 |
| 35 | Paritaprevir | −7.43470 |
| 36 | Peramivir | −6.7811 |
| 37 | Pibrentasvir | −9.87502 |
| 38 | Pimodivir | −6.79098 |
| 39 | Pleconaril | −7.16664 |
| 40 | Posaconazole | −7.95234 |
| 41 | Quinine | −6.67637 |
| 42 | Raltegravir | −7.76854 |
| 43 | Remdesivir | −5.9488 |
| 44 | Ribavirin | −6.3251 |
| 45 | Ribostamycin | −7.28802 |
| 46 | Rilpivirine | −7.09415 |
| 47 | Ritonavir | −9.94965 |
| 48 | Saquinavir | −9.25575 |
| 49 | Simeprevir | −9.50916 |
| 50 | Sofosbuvir | −8.4513 |
| 51 | Telaprevir | −8.96481 |
| 52 | Temsavir | −8.33862 |
| 53 | Tenofovir Diphosphate | −6.557 |
| 54 | Tipranavir | −7.47819 |
| 55 | Umifenovir | −7.23012 |
| 56 | Velpatasvir | −9.33025 |

**100 nanosecond (ns) Molecular Dynamics simulations for micafungin and ivermectin.** To investigate the stability of these docking poses, 100 ns molecular dynamics (MD) simulation studies were performed for two compounds, ivermectin and micafungin. Figure 7a panel I–III shows the MD simulations of micafungin with the monomer form of 3CLpro. The MD simulation data suggests that micafungin was stable and remained bound in the active site pocket throughout the 100ns simulated time (compare Fig. 7a I–III). The analysis of protein-ligand interaction fingerprints between the monomeric 3CLpro enzyme and micafungin (Fig. 7a, panel-IV) shows that micafungin has a predominant interaction with Glu[166]. It is possible that the micafungin remained bound in the pocket of the monomeric form of 3CLpro for the entire length of the 100-ns trajectory via a hydrogen bonding with Glu[166]. As noted earlier, interaction with Glu[166] suggests interference with the dimerization of the 3CLpro in SARS-CoV-2, which is required for its activity[28] thus explaining the inhibitory activity of micafungin against 3CLpro enzyme. Supplementary Movie 1 presents the real-time interaction of micafungin with the active site of monomeric 3CLpro indicating the stability of micafungin in the catalytic pocket of monomer.

The MD simulations for ivermectin are shown in Fig. 7b (panel II–III), where ivermectin diffuses out of the catalytic pocket of 3CLpro monomer after 85 ns. It is evident from the protein-ligand fingerprint map that ivermectin interacts with both Cys[145] and His[41] of the 3CLpro monomer for about 14 ns (Fig. 7b, panel-IV). Later, ivermectin loses its interaction with His[41] and does not show interaction with any amino acids of interest (Cys[145], His[41], and Glu[166]) and eventually diffuses out of the pocket at 85 ns (Fig. 7b, panel-II). Supplementary Movie 2 shows the instability of the ivermectin in the catalytic pocket of the monomeric form of 3CLpro. Since the homodimer is the active form of 3CLpro enzyme[28], we hypothesized that the homodimeric form of 3CLpro is required to stabilize ivermectin in the catalytic pocket and hence is responsible for the inhibitory activity of ivermectin. To test our hypothesis, MD simulations of ivermectin with the homodimer form of 3CLpro was performed. Interestingly, we observed that ivermectin remained bound in the catalytic pocket of the homodimer (compare Fig. 7c, panel I–III) throughout the period of the simulation. The detailed analysis of the homodimer 3CLpro-ivermectin fingerprint region (Fig. 7c, panel-IV) shows that ivermectin interacts with both Cys[145] and His[41] for 2 ns, then with His[41]. After 85 ns, ivermectin contacts with Ser[1] of the neighboring monomer, suggesting that this amino acid residue assists in the stabilization of ivermectin in the catalytic binding pocket (Fig. 7c, panel IV). Supplementary Movie 3 exhibits the real-time MD simulation interaction and the stability of ivermectin in the catalytic pocket with homodimer of 3CLpro from 0–100ns.

Further, Supplementary Fig. 3 shows the binding affinity (S-score) over the course of the MD simulation for ivermectin with the monomer and homodimer form of 3CLpro, and micafungin with monomer form of 3CLpro. We observed that the S-score for micafungin was stable over the period of computation whereas, ivermectin with monomer form of 3CLpro fluctuated from −9.64 to −2.2 kcal/mol. As shown in the Supplementary Fig. 3, upon leaving the active site ~85 ns ivermectin exhibited an increase in S-score after (~−2.0 kcal/mol). This is in stark contrast to the S-score of ivermectin in complex with the homodimer form of 3CLpro, which remained stable throughout the simulation with an average of −5.64 kcal/mol. Taken together, this computational model provides a framework for the possible interaction between these inhibitors and 3CLpro. However, the structural interaction of these drugs with SARS-CoV-2 3CLpro needs to be validated by X-ray crystallographic studies.

## Discussion

COVID-19 is a disease caused by the SARS-CoV-2 and is a major threat to public health globally because of the high rates of infection and mortality. Morbidity and mortality continue to rise

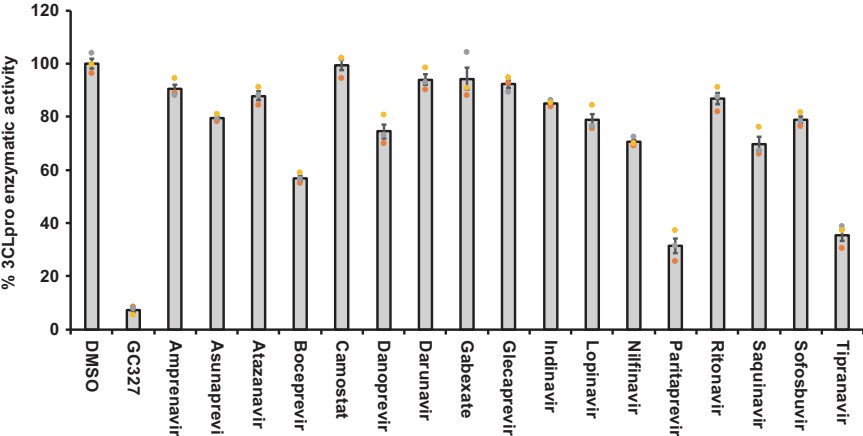

**Fig. 2 Protease inhibitors boceprevir, paritapravir, and tipranavir exhibited partial inhibitory activity against SARS-CoV-2 3CLpro enzyme.** The selected FDA approved viral-protease inhibitors were screened for their inhibitory activity against SARS-CoV-2 3CLpro enzyme as described under Methods section. The fluorescence units were converted to percent enzymatic activity considering DMSO treated control as 100% activity. Blank values were subtracted from all the readings before calculating the percent activity. Representative of three individual experiments with triplicate values were presented graphically ($n = 3$). P value < 0.001 considered as statistically significant. One-way ANOVA with Dunnett's Multiple Comparison post-hoc test was used to calculate the statistical significance.

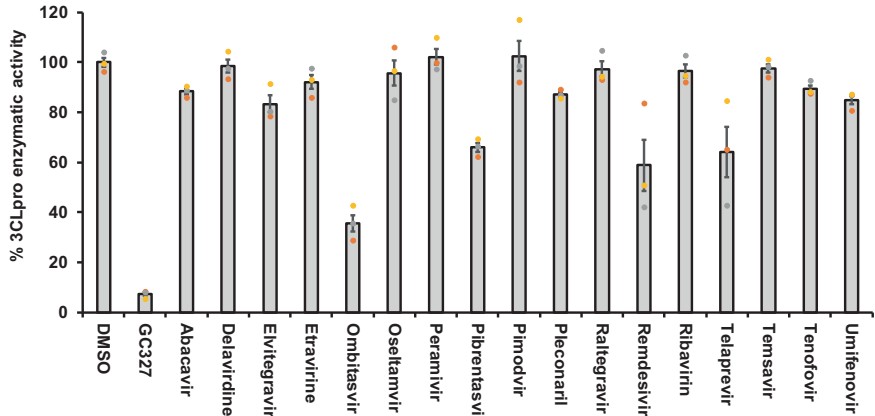

**Fig. 3 Non protease inhibitor ombitasvir inhibited SARS-CoV-2 3CLpro activity partially.** The non-protease anti-viral drugs selected by computational studies were screened for their inhibitory activity against SARS-CoV-2 3CLpro enzyme as described under Methods section. The percent enzymatic activity was calculated as described in Fig. 1 legend. Blank values were subtracted from all the readings before calculating the percent activity. Representative of three individual experiments with triplicate values were presented graphically ($n = 3$). P value < 0.001 considered as statistically significant. One-way ANOVA with Dunnett's Multiple Comparison post-hoc test used to calculate the statistical significance.

due to the lack of a specific vaccine and drugs that prevent COVID-19 disease progression. There is an urgent need to identify and test potential therapeutics for this disease. One approach that may lead to a more rapid increase in treatment options is to repurpose currently approved FDA drugs for their ability to prevent or reduce the spread of virus and severity of COVID-19 pathogenesis. Many of the FDA approved drugs are being repurposed in clinics to treat COVID-19. However, the effectiveness of these drugs and their specific targets for preventing or reducing the severity of symptoms of COVID-19 has not yet been completely established[2,29–31]. Therefore, several laboratories are identifying specific drugs for many targets of SARS-CoV-2. Herein, we have investigated 47 FDA approved drugs that inhibit the SARS-COV-2 3CLpro enzymatic activity, the main enzyme for viral replication and the preferred drug target for COVID-19[32]. We used MOE computational studies for the initial screening to select the drugs that have high affinity for 3CLpro and further functional inhibitory activity of the 47 selected drugs was confirmed using in vitro enzymatic assay. As noted in the previous studies[24,25], our data suggests that inhibitory

effects of drugs predicted from the computational screening as defined by the S-score do not agree with our experimental in vitro studies. Thus, additional in vitro screening for all the drugs is warranted.

Among 17 PIs tested here, boceprevir, paritaprevir, and tipranavir were able to partially inhibit the 3CLpro enzymatic activity at 50 μM drug concentration (Fig. 2). It has been shown that boceprevir and paritaprevir inhibit the hepatitis-C virus by inhibiting protease activity of nonstructural protein 3 and 4A (NS3/4A)[33,34]. Tipranavir is a retroviral protease inhibitor that binds to active site of HIV protease and prevents proteolytic cleaving of precursor polyproteins into mature functional proteins thereby inhibiting the viral replication[35]. Apart from these drugs, known PIs lopinavir and ritonavir did not exhibit any 3CLpro inhibitory activity (Fig. 2); thus explaining the ineffectiveness of these drugs in clinical trials of COVID-19 treatment[31]. Our observation with lopinavir and ritonavir are in agreement with Ma et al.[34]. Further, the VNI agent such as ombitasvir which targets the nonstructural proteins (NS5A) of hepatitis-C virus to inhibit the viral replication and assembly[36] was also able to

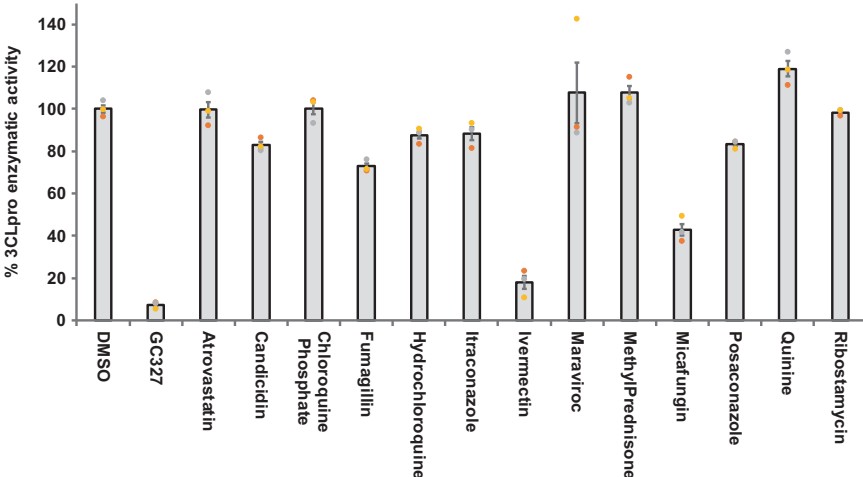

**Fig. 4 Ivermectin exhibited complete inhibition of SARS-CoV-2 3CLpro enzymatic activity whereas micafungin partially inhibited the enzyme.** The off-target drugs that are being used to treat non-viral ailments selected by in silico studies were screened for their inhibitory activity against SARS-CoV-2 3CLpro enzyme as described under Methods section. The percent enzymatic activity was calculated as described in Fig. 1 legend. Blank values were subtracted from all the readings before calculating the percent activity. Representative of three individual experiments with triplicate values were presented graphically ($n = 3$). $P$ value < 0.001 considered as statistically significant. One-way ANOVA with Dunnett's Multiple Comparison post-hoc test used to calculate the statistical significance.

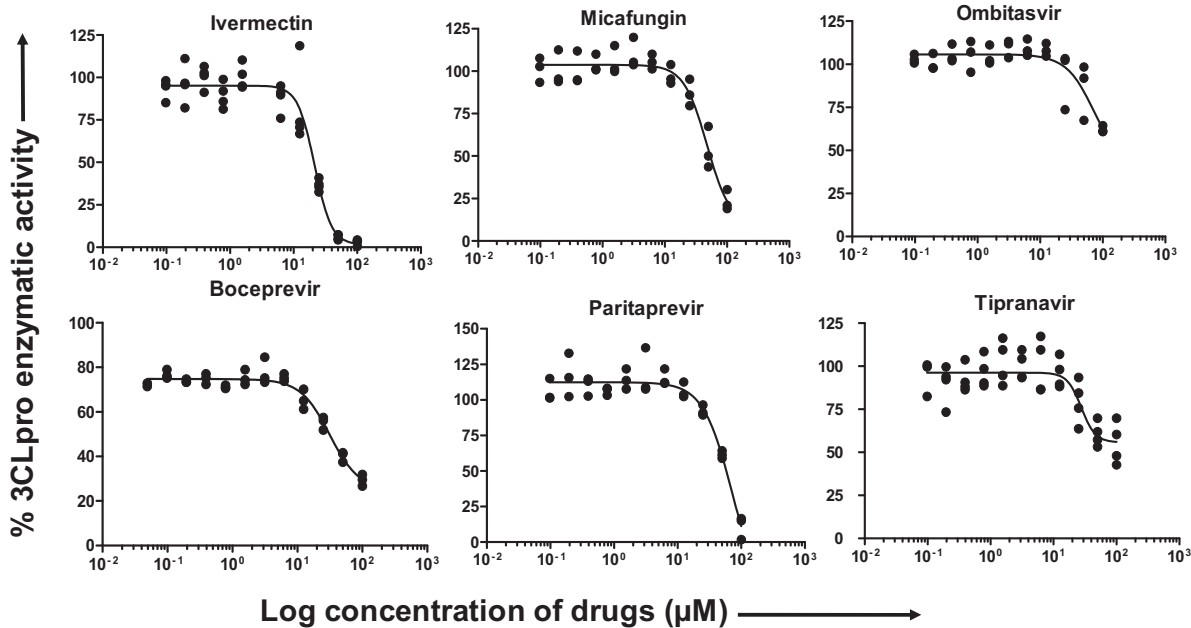

**Fig. 5 Dose-dependent inhibition of SARS-CoV-2 3CLpro activity by selected PIs, VNIs, and OTDs.** The drug candidates that exhibited more than 50% of inhibitory activity at 50 µM concentration were selected for dose-dependent and $IC_{50}$ calculation studies. A serial dilution of drugs ranging from 0 to 100 µM in assay buffer was used. The percent activity was calculated as described in Fig. 1 legend. Representative of three individual experiments with triplicate values were presented graphically ($n = 3$). Non-linear regression (curve fit) with four variable dose vs inhibition was used to calculate the $IC_{50}$ values using GraphPad Prism.

**Table 2 $IC_{50}$ values of non-viral protease inhibitor drugs for SARS-CoV-2 3CL protease.**

| SL. No. | Drug name | $IC_{50}$ values |
|---|---|---|
| 1 | Ivermectin | 21.53 |
| 2 | Tipranavir | 27.66 |
| 3 | Boceprevir | 31.36 |
| 4 | Micafungin | 47.63 |
| 5 | Paritaprevir | 73.38 |
| 6 | Ombitasvir | 75.49 |

partially inhibit the 3CLpro enzymatic activity in vitro. In addition to the anti-viral drugs, micafungin, one of the OTDs tested, also inhibited 66% of 3CLpro activity. Micafungin is a broad spectrum anti-fungal drug which belongs to the class of echinocandin and acts by targeting fungal β-1-3 glucan synthase[37].

While the reason for the partial inhibitory effect of the agents boceprevir, ombitasvir, paritaprevir, tipranavir, and micafungin towards 3CLpro is not clearly understood, it is possible that the strength of the hydrogen bonding interactions between these agents and Cys[145]/His[41]/Glu[166] of 3CLpro may explain differential inhibitory effect. Nonetheless, our studies provide avenues

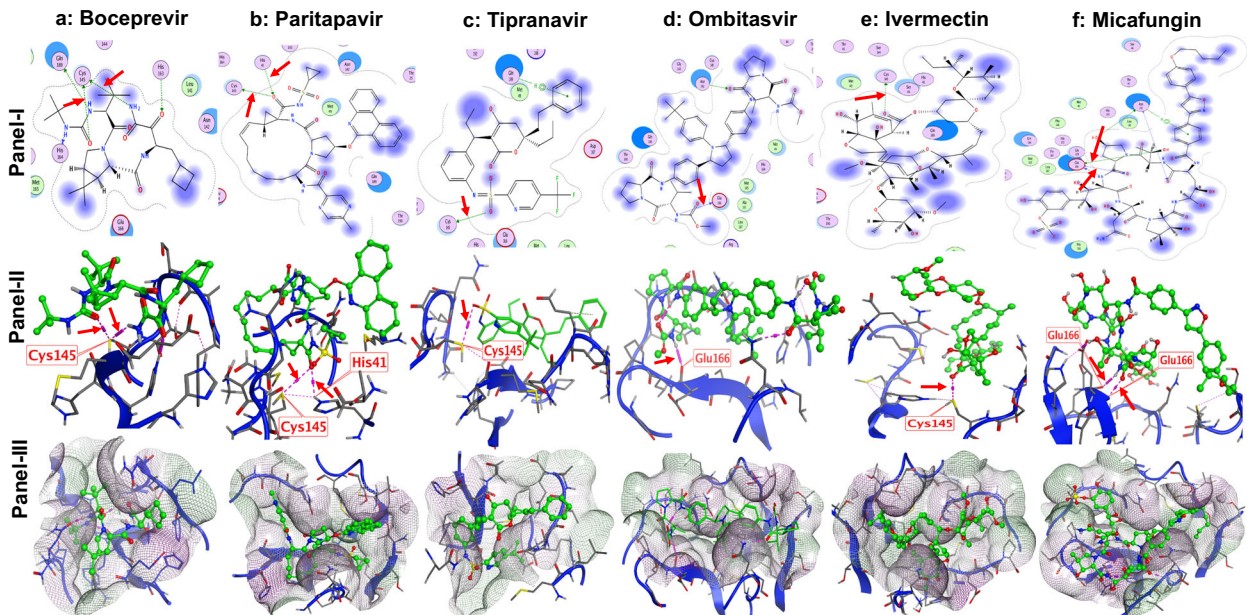

**Fig. 6 Structural analysis and interaction of the potential inhibitors with SARS-CoV-2 3CLpro enzyme.** Inhibitors were docked with active site of 3CLpro protein. Panel-I: ligand interaction map. Panel-II: interaction of inhibitors specific amino acids of 3CLpro at active site. Panel-III: lipophilic cavity of active site with drugs interacting with specific amino acids. Boceprevir (**a**), partapravir (**b**), tipranavir (**c**), ombitasvir (**d**), ivermectin (**e**), and micafungin (**f**) are arranged in columns for comparison. Drugs are represented in green color with ball and stick model. Arrows indicate the C–H, N–H, and C–O bonds between drugs and with Cys[145], His[41], and Glu[166] residues since they are essential for the enzymatic activity of 3CLpro enzyme. However, we also observed the drugs interacting with neighboring amino acid residues.

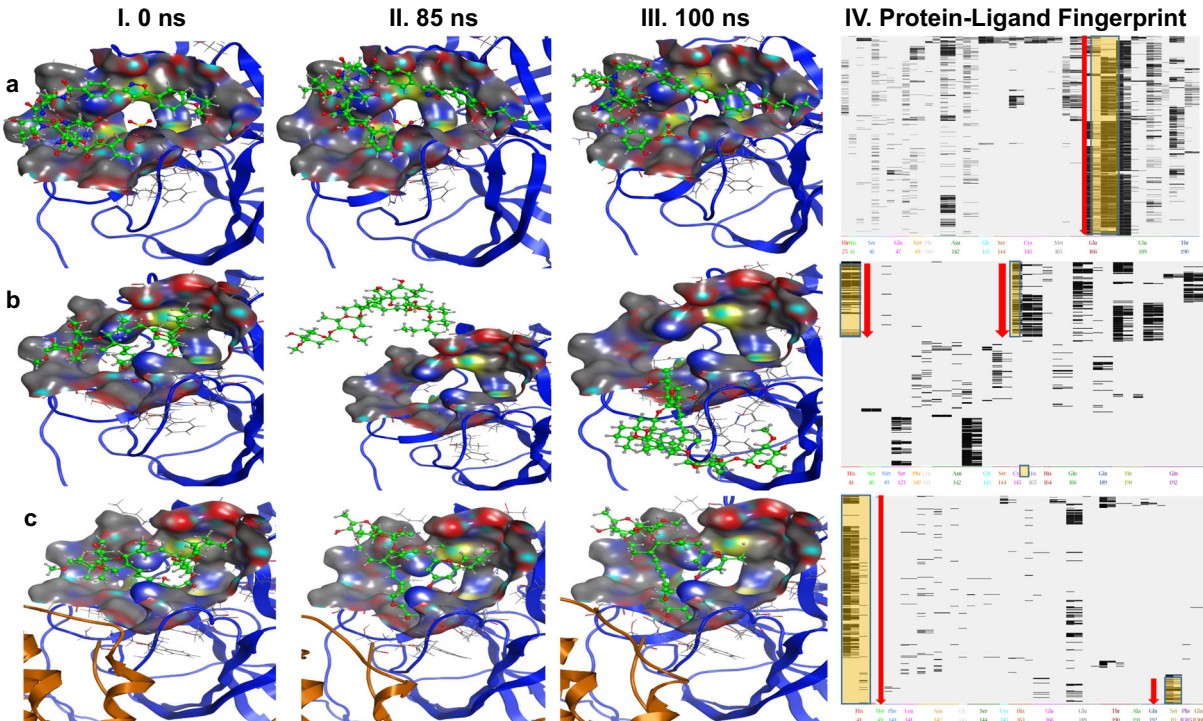

**Fig. 7 MD 100 ns simulation studies for ivermectin and micafungin.** MD simulation studies were carried as described under Methods section. Interaction of micafungin (**a**), and ivermectin (**b**) with monomeric form of 3 CLpro enzyme. **c** Interaction of ivermectin with active site of 3CLpro homodimer. Panel I–III represents the interaction of ligand at different time points in nano second (ns). Panel-IV represents the ligand-binding fingerprint of micafungin and ivermectin with specific amino acids of 3CLpro enzymes.

to use the scaffold of these molecules to design highly potent and specific inhibitors for SARS-CoV-2 3CLpro enzyme.

Interestingly, one of the OTD, ivermectin was able to inhibit more than 85% (almost completely) of 3CLpro activity in our

in vitro enzymatic assay with an IC$_{50}$ value of 21 μM. These findings suggest the potential of ivermectin to inhibit the SARS-CoV-2 replication. In support of this, a recent finding suggested that ivermectin (5 μM) inhibited the replication of live

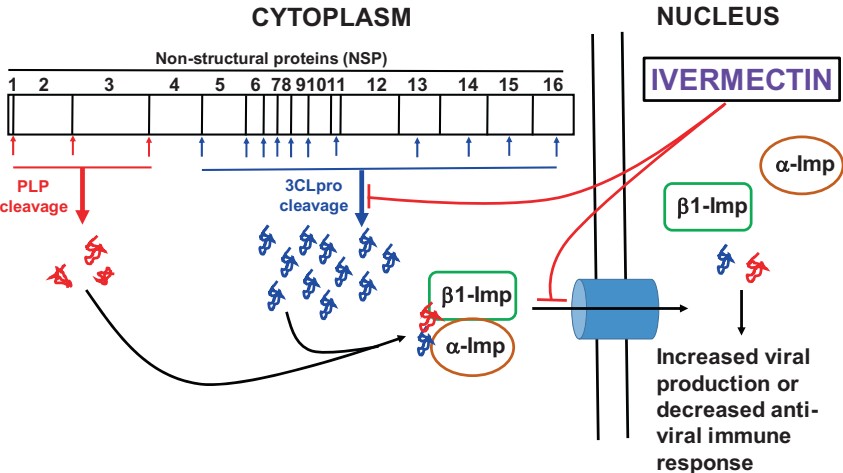

**Fig. 8 Schematic representation of anti-viral mechanism of ivermectin.** Hypothetical model illustrating the inhibition of SARS-CoV-2 replication by ivermectin mediated through the blocking of α/β1-importin (imp) as well as 3CLpro enzymatic activity.

SARS-CoV-2 isolated from Australia (VIo1/2020) in Vero/ hSLAM cells[23]. They found that >5000-fold viral counts were reduced in 48 hr in both culture supernatant (release of new virion: 93%) as well as inside the cells (unreleased and unassembled virion: 99.8%) when compared to DMSO treated infected cells. Interestingly, this study reported the $IC_{50}$ value of ivermectin as 2.5 μM[23], whereas, we observed an $IC_{50}$ value of 21 μM (10-fold increase). The variability in the $IC_{50}$ values reported could be attributed towards the differences in the assay conditions such as the use of live virus vs enzymatic assay with purified 3CLpro protein. Further, preclinical studies need to be established to validate the in vivo inhibitory activity and $IC_{50}$ values of ivermectin.

Ivermectin is known to be effective against many positive-sense, single stranded RNA viruses such as Zika, Dengue, Yellow fever, West nile, Venezuelan equine encephalitis, Chikungunya, Semliki forest, Sindbis, Rorcine Reproductive and Respiratory Syndrome, and Human immunodeficiency-1 viruses[38]. The list of anti-viral effects of ivermectin against other RNA and DNA viruses were summarized in a recent review[38]. Earlier studies have demonstrated that the possible anti-viral mechanism of ivermectin was through the blockage of viral-protein transportation to the nucleus by inhibiting the interaction between viral protein and α/β1 importin heterodimer, a known transporter of viral proteins to the nucleus especially for RNA viruses[19–23]. However, in this study, we have reported that ivermectin inhibits the enzymatic activity of SARS-CoV-2 3CLpro and thus may potentially inhibit the replication of RNA viruses including SARS-CoV-2. These studies suggest that ivermectin could be a potential drug candidate to inhibit the SARS-CoV-2 replication and the proposed anti-viral mechanism of ivermectin presented in Fig. 8 and in vivo efficacy of ivermectin towards COVID-19 is currently been evaluated in clinical trials (ClinicalTrials.gov Identifier: NCT04438850).

In conclusion, the SARS-CoV-2 specific 3CLpro enzyme was used as a target to screen the potential drugs that have high binding affinity for 3CLpro since it plays a major role during viral replication. We have identified that boceprevir, micafungin, ombitasvir, paritaprevir, and tipranavir exhibited partial inhibitory effect, whereas ivermectin was able to completely inhibit the SARS-COV-2 3CLpro enzymatic activity in vitro at the tested doses. The 100 ns MD simulation studies suggest that the ivermectin may require homodimeric form of 3CLpro enzyme for its inhibitory activity. This could be due to the interaction of the amino acid residue, Ser[1], from the neighboring monomer. On the

other hand, micafungin remained bound in the catalytic pocket of the monomeric from of 3CLpro throughout the period of simulation. The list of drugs that are reported in this study provides a rationale to prioritize these potential drugs to be tested preclinically followed by clinical studies to target the SARS-Cov-2 pathogenesis.

## Methods

**Reagents and drugs**. Molecular biology grade DMSO was purchased from Sigma Aldrich (St. Louis, MO, USA), Sterile PBS was from ThermoFisher Scientific (Waltham, MA, USA). 3CLpro inhibitor screening enzymatic assay kits (Catalog #79955-1) were from BPS Biosciences (San Diego, CA, USA). Amprenavir, Atazanavir sulfate, Candicin, Chloroquine Phosphate, Hydroychloroquine Sulfate, and Lopinavir were purchased from Sigma Aldrich (Saint Louis, MO). Beclabuvir, Temsavir were from Medchem Express (Monmouth Junction, NJ). Abacavir (sulfate), Arbidol hydrochloride, Asunaprevir, Atrovastatin, Boceprevir, Daclatasvir, Danoprevir, Darunavir, Delavirdine (mesylate), Edoxudine, Elbasvir, Elvitegravir, Etravirine, Favipiravir, Fumagillin, Glecaprevir, Grazoprevir, Indinavir sulfate, Itraconazole, Ivermectin, Ledipasvir (G-5885), Maraviroc, Methylprednisolone, micafungin sodium, ombitasvir, Oseltamivir phosphate, Paritaprevir, Peramivir, Pibrentasvir, Pimodivir, Pleconaril, Posaconazole, Quinine, Raltegravir (potassium salt), Remdesivir, Ribavirin 5'-monophosphate (lithium salt), Ribostamycin sulfate, Rilpivirine, Saquinavir, Telaprevir, Tenofovir diphosphate (sodium salt), and Velpatasvir were purchased from Cayman Chemicals (Ann Arbor, MI).

**Preparation of working solution of ligands**. One mg of drug was used to prepare 8 mM stock solution using either DMSO or PBS as solvent. This stock solution was used to prepare the working solution of 250 μM and 500 μM of drugs in PBS.

**3CLpro protein preparation**. The crystal structure of COVID-19 main protease was retrieved from the protein data bank (www.rcsb.org) with PDB format (ID: 6LU7). Any structural issues if present in the protein were corrected using QuickPrep option in MOE. The QuickPrep option performs a protonation and calculates the minimum energy conformation of the protein. Default parameters of MOE software was used for QuickPrep function.

**Preparation of ligands**. The drug list of 3987 FDA approved drug molecules and active ingredients were downloaded from SUPERDRUGS2 database in sdf format. The database of these 3987 drugs were imported into Molecular Operating Environment MOE and were cleaned using wash function in MOE. This function rebalances protonation states and regenerate 3D coordinates to their minimum energy conformations. Default parameters of MOE software was used for wash function.

**Protein:drug docking studies**. Integrated Computer-Aided Molecular design computing method Molecular Operating Environment (MOE) software was used to dock the drugs with 3CLpro protein. Briefly, docking was performed on the ligand site of 6LU7 protease using washed dataset of 3987 drugs. 30 poses for London dG and 20 poses of GBVI/WSA dG were used for final docking. All docking results were sorted by the binding energy using S Score function and they were viewed for accuracy using the ligand interaction function in MOE. The drugs with a binding score (S-Score) of ≤−6.5 were considered for further studies. The

list was further narrowed down based on the listed criteria (1) All peptidomimetic drugs were eliminated as they tend to elicit immune response in the body; (2) Drugs used as immunosuppressants or for the treatment of cardiovascular disorders were eliminated as they tend to be in high risk categories; (3) All prodrugs were eliminated unless the active form was readily available as our in vitro assay is specific for SARS-COV-2 3CLpro and does not provide an option to activate drugs in vitro; and (4) Drugs with a shorter half-life of <30 min were also eliminated. In addition, we also ensured that small molecules which are part of any current Phase I, II, or III clinical trials were included in this study irrespective of their S-Score. For e.g favipiravir has an S score of −4.23 and since it is under clinical trials for COVID-19, we have included favipiravir in the study. This screening led to the database of 56 drugs, which were selected to study their inhibitory effect on SARS-CoV-2 specific 3CLpro enzyme using an in vitro 3CLpro enzymatic assay.

**3CLpro enzymatic assay**. SARS-CoV-2 specific 3CLpro assay kits were purchased from BPS Biosciences (CA) and enzymatic assay was carried out as per the manufacturer's protocol using 96 well plates. Briefly, 4 ng 3CLpro-MBP tagged enzyme in 30 μl of assay buffer was pre-incubated with 10 μl of (250 μM) drugs for 1 h. The enzymatic reaction was initiated by adding 10 μl (250 μM) fluorescent substrate. The final volume of the assay samples was 50 μl. The final concentration of drugs and substrate in the reaction mixture was 50 μM. Incubation was continued at room temperature for 16–18 h. Fluorescence reading was taken at 360/40 excitation and 460/40 nm emission using Synergy HT fluorescent plate reader. For $IC_{50}$ calculation, drugs were screened from 0 to 100 μM dose range. Wells with 1% DMSO with 4 ng of enzyme and 50 μM of substrate served as positive control with no enzyme inhibition. Wells with 50 μM of GC367 compound (provided by the BPS Biosciences) served as standard inhibitor and negative control. Wells with 1% DMSO with 50 μM of substrate without enzyme served as blank. All the values were subtracted from blank values.

**Molecular dynamic simulation studies**. The structure of COVID-19 main protease 6LU7 (PDB ID: 6LU7) was prepared in its monomeric and functional dimeric form using the QuickPrep application of MOE2019 with default parameters. This atomistic model was used for generating the input files for all MD simulations.

The best dock pose of micafungin was placed into the binding pocket of the prepared monomeric 3CLprotease structure. The best pose of ivermectin was placed into the monomeric and dimeric protease structure to test which form can stabilize the inhibitor. The simulation cell and NAMD 2.14[39] input files were generated using MOE2019. The crystallographic water molecules were removed prior to solvation. Next, the protein/ligand complexes were embedded in a TIP3P water box with cubic periodic boundary conditions, keeping a distance of 10 Å between the boundaries and the protein. The net charge of the protein was neutralized with 100 mM NaCl. For energy minimization and MD simulations, the AMBER10:EHT force field was used and the electrostatic interactions were evaluated by the particle-mesh Ewald method. Each system was energy-minimized for 5000 steps using the Steepest Descent and Conjugate Gradient method. For equilibration the system was subjected to a 100-ps simulation to gradually heating the system from 10 K to 300 K. Next, a 100-ps NVT ensemble was generated at 300 K followed by an NPT ensemble for 200 ps at 300 K and 1 bar. Then, for each complex, a 100-ns production trajectory was generated for further analysis. The trajectory analysis was done using scripts shared by the CCG support group.

**Identification of ligand-binding mode**. The protein ligand interaction fingerprint application in MOE2019 was used to study the average binding mode of each inhibitor bound to the monomer or dimer of 6LU7. The calculation used the default parameters on the recorded MD trajectories.

**Statistics and reproducibility**. One-way analysis of variance (ANOVA) with Dunnett's Multiple Comparison post-hoc test was performed with 99.9% confidence intervals to compare the statistical significance and represented as the mean ± SEM. $P$ values < 0.001 considered statistically significant. Non-linear regression (curve fit) with four variable dose vs inhibition was used to calculate the $IC_{50}$ values. Statistical analysis was performed using GraphPad Prism (version 6.07; La Jolla, CA, USA). All the experiments were carried out minimum three times with triplicates for reproducibility and the representative of three individual experiments is presented in this report. The data generated at different time points were combined to make the final graphs. Investigators performing the assay were blinded for the drugs being tested in the assay.

**Reporting summary**. Further information on research design is available in the Nature Research Reporting Summary linked to this article.

## Data availability

Supplementary Data 1 provides the data set for Figs. 2–5, and Supplementary Figure 2 and 3. Supplementary Data 2 and 3 provides data set for 3CLPro-OTDs docking study and Supplementary Data 4 for PIs and VNIs (S score for Table 1 and structural analysis for Fig. 6). Supplementary Data 5 provides data set for MD simulation study of 3CLPro homodimer with ivermectin, Supplementary Data 6 for 3CLPro monomer with

ivermectin and Supplementary Data 7 for 3CLPro monomer with micafungin (Fig. 7). Supplementary Data 5–7 provides data set for Supplementary Fig. 3 (S score comparison from MD simulation study). Any remaining information can be obtained from the corresponding author upon reasonable request.

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

## Acknowledgements

We thank Travis Field, Tu Dang, Handong Ma, and Yan Wu, Division of Research, PCOM for their technical support during the study. We also thank Dr. Deepa Machiah, Emory University, Atlanta for critical inputs and proof reading the manuscript. This work was supported by Chief Scientific officer funding from PCOM to V.M. and S.T., and partly by National Institute of Allergy and Infectious Diseases (1R03 AI128254-01A1) to S.T.

## Author contributions

V.M. and S.T. designed, performed and wrote the manuscript, S.R. performed the experiment and proofread the manuscript. M.E. performed MD simulation studies. G.M.F. helped with molecular docking studies. J.H., S.W., A.M., and L.L. collected the data on drugs. All reviewed the manuscript.

## Competing interests

The authors declare no competing interests.
