## [Peer Review File · Communications Biology]

Reviewers' comments:

Reviewer #1 (Remarks to the Author):

The manuscript titled "Identification of potential inhibitors for SAR-CoV-2 3CLpro enzyme" addresses to check if Mpro enzyme was inhibited with 3987 known FDA approved drugs. They found that 48 of them are active against 3CLpro protease of COVID-19. The authors tested these drugs experimentally toward 3CLpro enzyme and it was found that off-target inhibitors such as ivermectin and micafungin inhibited more than 85 and 66% of 3CLpro enzyme activity, respectively. It is original, nice flow and includes both experimental and computational studies of the potential drug candidates. I believe the manuscript is timely and will get the attention of many researchers studying in this field. The paper is publishable taking into consideration of the following comments:

Comments:

1. It is important to know how the computational inhibition constants (or binding energies) agrees the experimental result. Is there any correlation between them?
2. 100 nano second MD simulation is recommended for at least the best 2 drug candidates like ivermectin and micafungin to follow their stability and dynamic behavior in the active sites.
3. Some of the pictures (2D and 3D) are not clear at all and not readable. Nature of the interactions are very important in the binding site and the color designation and type of interactions should be provided.

Reviewer #2 (Remarks to the Author):

This work, based on the molecular docking and cell-free biological inhibition tests, suggested several FDA approved-drugs having the potential to directly inhibit 3CLpro, one key enzyme for SARS Cov-2 to replicate.

There are two concerns about this work.

1. The conflict of the molecular docking results and ELISA results. The author used two steps to screen drugs, the first step is to screen by molecular docking, and only molecules whose binding energy between molecules and proteins are below a certain threshold are retained. In the second step, an in vitro ELISA experiment was used to determine the actual inhibition of the enzyme by the drugs screened in the first step. The implicit idea of this two-step screening protocol is that the results of the first step should be consistent with the results of the second step, that is to say, the drugs predicted to bind to the enzyme in the first step should be displayed a practical inhibitory effect in the second step. However, of the 48 drugs that passed the first step screening, only 7 showed inhibitory effects in the second step, the majority (41/48) are not inhibitory. In addition, in those 7 drugs showing inhibition, their inhibitory ability are not consistent with the binding ability predicted in the first screening. For example, the predicted binding capacity of Ivermectin (-7.74053) to the enzyme is weaker than that of Ombitasvir (-8.96571), but in the ELISA experiment, the actual inhibition of the enzyme by Ivermectin ($IC_{50}=21.53\mu M$) is stronger than that of Ombitasvir ($IC_{50}=75.49\mu M$). Therefore, the predications in the first step are not consistent with the results of the second step. To this end, why is necessary to perform the first step prediction? More seriously, does the first prediction step wrongly expel inhibitory drugs? The author should explain this. In addition, if possible, the author could randomly select several drugs

from those excluded in the first prediction and perform ELISA experiments on these drugs to observe whether these drugs also have inhibitory effects. If none of the excluded drugs show inhibitory effects, the first step of screening is meaningful.

2. The concentration of the drug to exert effective inhibition. The inhibitory concentrations of the drugs selected by the author are all at the μM level. Can such concentrations be achievable when these drugs are used in clinical? If possible, the author should provide the maximum plasma concentration of the screened drug based on the reported data.

Reviewer #3 (Remarks to the Author):

The work "Identification of 3-chymotrypsin like protease (3CLPro) inhibitors as potential anti-SARS-CoV-2 agents" by Taval et al. is an interesting exercise in the combined use of molecular docking and functional in vitro assays to quickly screen known drug compounds for potential re-purposing against a new target, specifically the main protease (3CLPro) of the SARS CoV-2 virus.

Overall, the work is interesting, though some details and additional analysis should be provided for readers. Below are several comments that the authors should reflect upon.

0) When docking, how large of a search space did the authors use? Using a highly restricted search space could provide erroneously the high-ranking of some compounds which could, in reality, be binding to the protein away from the active site. It would be useful to take the top $\sim 1.5\%$ compounds (the 57 identified initially) and re-run the docking 'blind', i.e. with a search box that encompasses the entire protein and see which proportion are consistently binding to the enzyme active site.

1) In the discussion the authors note that both boceprevir and paritaprevir inhibit the hepatitis-C virus. Do the authors have a quantification of the structural similarity between the 3CLpro (MPro) of SARS-CoV-2 and the NSP3 and 4A of HepC? Could the inhibition mechanism be similar for the drug to both viruses?

2) It is unclear from the methods: was water present during the docking? This should be clarified so others can repeat the docking for their own benchmarks.

3) An additional method question: which protonation (delta or epsilon) state was assigned to His41?

4) The range of scores reported for the top 57 compounds is quite wide; could the authors provide a figure showing the distribution of all scores and provide justification for their score-based cutoff for experimental testing? It is unclear why the authors choose their specific score cutoff.

5) The authors have identified several compounds with inhibitory effects; it would be of use for the authors to determine and report upon the chemical similarity of each of the inhibitory compounds to address the open question of whether or not there exists a common 'molecular scaffold' that could be used in future drug development, beyond drug re-purposing, for SARS CoV-2

We thank our reviewers for the insightful comments, all of which are addressed below and/or in the revised manuscript. Changes to the text are highlighted in blue font in the revised manuscript.

Reviewer 1 Concerns

- 1. Comment:** It is important to know how the computational inhibition constants (or binding energies) agrees the experimental result. Is there any correlation between them?

Response: The protein structure-based drug design with the use of computational method is an alternative for screening currently approved drugs to rapidly identify potential drug candidates for the treatment of novel infectious diseases such as COVID-19. However, the potential for false positives with the predicted binding energy is one of the most common limitation of docking studies. In the current study, we did not find any correlation between the *in vitro* results of selected drugs and their computational inhibition constants. However, computational studies are being widely used to predict the initial protein-drug interactions. We believe additional *in vitro* screening of the drugs is necessary to confirm the inhibitory activities of the drugs.

We have included the above sentences in the revised manuscript for better readership.

- 2. Comment:** 100 nano second MD simulation is recommended for at least the best 2 drug candidates like ivermectin and micafungin to follow their stability and dynamic behavior in the active sites.

Response: We appreciate this valuable suggestion. We have performed the 100 nano second MD simulation in collaboration with Chemical Computing Group, 910-1010 Sherbrooke W, Montreal, Canada and included the results in the revised manuscript. We believe the MD data significantly improved our manuscript.

- 3. Comment:** Some of the pictures (2D and 3D) are not clear at all and not readable. Nature of the interactions are very important in the binding site and the color designation and type of interactions should be provided.

Response: We apologize for the quality of the images. We have included the high-resolution 2D and 3D pictures for the structural analysis and MD simulation studies and changed the background of the pictures to lighter color to make the binding sites/bonds visible to the readers. We have also included the description in each of the figure as suggested.

Review 2 Concerns

- 1. Comment:** The conflict of the molecular docking results and ELISA results. The author used two steps to screen drugs, the first step is to screen by molecular docking, and only molecules whose binding energy between molecules and proteins are below a certain threshold are retained. In the second step, an *in vitro* ELISA experiment was used to determine the actual inhibition of the enzyme by the drugs screened in the first step. The implicit idea of this two-step screening protocol is that the results of the first step should be consistent with the results of the second step, that is to say, the drugs predicted to bind to the enzyme in the first step should be displayed a practical inhibitory effect in the second step. However, of the 48 drugs that passed the first step screening, only 7 showed inhibitory effects in the second step, the majority (41/48) are not inhibitory. In addition, in those 7 drugs showing inhibition, their inhibitory ability are not consistent with the binding ability predicted in the first screening. For example, the predicted binding capacity of Ivermectin (-7.74053) to the enzyme is weaker than that of Ombitasvir (-8.96571), but in the ELISA experiment, the actual inhibition of the enzyme by Ivermectin (IC₅₀=21.53μM) is stronger than that of Ombitasvir (IC₅₀=75.49μM). Therefore, the predications in the first step are not consistent with the results of the second step.

To this end, why is necessary to perform the first step prediction? More seriously, does the first prediction step wrongly expel inhibitory drugs? The author should explain this. In addition, if possible, the author could randomly select several drugs from those excluded in the first prediction and perform ELISA experiments on these drugs to observe whether these drugs also have inhibitory effects. If none of the excluded drugs shows inhibitory effects, the first step of screening is meaningful.

Response: We agree with the reviewer and we did not find any correlation between the computational binding score and their *in vitro* activity we have included the following sentences in the revised manuscript to make better readership regarding the limitations of the computational studies.

“The protein structure-based drug design using computational methods is an alternative for screening of currently approved drugs to rapidly identify potential drug candidates for the treatment of novel infectious diseases such as COVID-19. However, the potential for false positives with the predicted binding energy is one of the most common limitation of the docking studies. In the current study, we did not find any correlation between the *in vitro* results of selected drugs and their computational inhibition constants. Therefore we believe additional *in vitro* screening of the drugs is necessary to confirm the inhibitory activities of the drugs”

We also agree with the reviewer that cannot rule out the possibility that we might have unintentionally eliminated some of the inhibitory drugs out of the 3,930 remaining drugs screened. However, under the current circumstances, it is difficult to screen thousands of drugs *in vitro*, which is very tedious and labor intensive. Therefore, for rapid screening, we used computational studies as a tool to select the drugs that could have potential inhibitory effect and further verified by conducting *in vitro* studies. Additionally, some of the predicted drugs are also in clinical trials and hence we believe that the data presented in our manuscript will be useful for the scientific community. Hence, we believe that selecting a few drugs randomly from those excluded in the first prediction to perform the *in vitro* studies will not add further value to the study.

- 2. Comment:** The concentration of the drug to exert effective inhibition. The inhibitory concentrations of the drugs selected by the author are all at the μM level. Can such concentrations be achievable when these drugs are used in clinical? If possible, the author should provide the maximum plasma concentration of the screened drug based on the reported data.

Response: Thank you for the suggestion, the IC_{50} values reported in this manuscript are based on the *in vitro* enzymatic studies at the tested dose of the drugs with fixed concentration of the enzyme. However, the IC_{50} values may vary with the live virus challenge in the cell based assays or *in vivo* studies. Since our manuscript details more on the basic mechanistic studies and *in vitro* inhibitory activities of certain drugs towards SARS-CoV-2 3CLpro enzyme, we believe including the plasma concentration, $t_{1/2}$ and *in vivo* dose may not be relevant and may add to further confusion. Moreover, these parameters change based on the dose and route of administration. For example, ivermectin plasma concentration varies significantly when administered orally vs subcutaneously (PMID: 18446504). Therefore, we did not include this information in the revised manuscript.

Review 3 Concerns

- 1. Comment:** When docking, how large of a search space did the authors use? Using a highly restricted search space could provide erroneously the high-ranking of some compounds which could, in reality, be binding to the protein away from the active site. It would be useful to take the top ~1.5% compounds (the 57 identified initially) and re-run the docking 'blind', i.e. with a search box that encompasses the entire protein and see which proportion are consistently binding to the enzyme active site.

Response: We appreciate the suggestion. We performed the site-specific docking studies, as it is the preferred method to have better docking results when the crystal structure of the protein with inhibitor is known. Therefore, we used the crystal structure of 3CLpro with inhibitor at the active site and for docking studies, we replaced the inhibitor with drugs and calculated the binding energy. However, we also performed the blind docking studies as suggested but we did not find any correlation between the in vitro activity of the drugs and their S scores as determined computationally (data not show).

- 2. Comment:** In the discussion the authors note that both boceprevir and paritaprevir inhibit the hepatitis-C virus. Do the authors have a quantification of the structural similarity between the 3CLpro (MPro) of SARS-CoV-2 and the NSP3 and 4A of HepC? Could the inhibition mechanism be similar for the drug to both viruses

Response: We have compared the protein sequence of NSP3/4A complex (PDB: 1A1R_A) of HepC and 3CLpro of SARS-CoV-2 using NCBI protein sequence BLAST program. There is only 30% sequence homology and we did not find the structural similarity in the catalytic site with Cys¹⁴⁵-His⁴¹ dyad, which is a pre-requisite for 3CLpro activity. Therefore, we believe that the interaction could be due to the molecular dynamics of the drugs.

- 3. Comment:** It is unclear from the methods: was water present during the docking? This should be clarified so others can repeat the docking for their own benchmarks.

Reponses: We docked with and without water molecules and did not find any difference in the binding energy; therefore, we removed the water molecule from our docking studies.

- 4. Comment:** An additional method question: which protonation (delta or epsilon) state was assigned to His41?

Response: We used Quick Prep option in MOE to automatically protonate residues and we have not manually assigned any protonation states. The protein preparation without any ligand revealed that epsilon protonation is dominant in histidine 41.

5. **Comment:** The range of scores reported for the top 57 compounds is quite wide; could the authors provide a figure showing the distribution of all scores and provide justification for their score-based cutoff for experimental testing? It is unclear why the authors choose their specific score cutoff.

Response: The drugs with a binding score (S-Score) of ≤ -6.5 were considered for our studies. Thalluri et al (2020; PMID: 32798373) studies used -7 as cutoff so we used -6.5 to include few more additional drugs. This information and the reference are included in the revised manuscript.

6. **Comment:** The authors have identified several compounds with inhibitory effects; it would be of use for the authors to determine and report upon the chemical similarity of each of the inhibitory compounds to address the open question of whether or not there exists a common 'molecular scaffold' that could be used in future drug development, beyond drug re-purposing, for SARS CoV-2.

Response: Thank you for the suggestions. We have compared the structure of all the molecules exhibiting inhibitory effect, however, we did not find a common scaffold among these molecules. We believe that the scaffold of each molecules should be considered separately for future drug development.

REVIEWERS' COMMENTS:

Reviewer #1 (Remarks to the Author):

Thank you very much all the points that I raised were corrected.

Reviewer #2 (Remarks to the Author):

The authors have addressed my concerns. There are no other questions.

Reviewer #3 (Remarks to the Author):

The revised manuscript "Identification of 3-chymotrypsin like protease (3CLPro) inhibitors as potential anti-SARS-CoV-2 agents" is an improvement over the original submission and many of my concerns have been addressed.

The use of MD, while very useful, appears to be more cursory than truly informative. A single 100ns MD simulation doesn't actually tell you anything about binding strength nor does it say much of the binding kinetics as it is a single calculation. The authors should be careful in noting this limitation.

Other than that one comment, the manuscript appears to have addressed my concerns.